# ab-Initio Study of Hydrogen Bond Networks in 1,2,3-Triazole Phases

**DOI:** 10.3390/molecules25235722

**Published:** 2020-12-03

**Authors:** Christopher Peschel, Christian Dreßler, Daniel Sebastiani

**Affiliations:** Institute of Chemistry, Martin-Luther-Universität Halle-Wittenberg, von-Danckelmann-Platz 4, 06120 Halle, Germany; christopher.peschel@chemie.uni-halle.de (C.P.); christian.dressler@chemie.uni-halle.de (C.D.)

**Keywords:** ab-initio molecular dynamics, proton conductivity, crystal structure, hydrogen bond network, tautomerism, fuel cell, pi-pi-stacking, diffusion coefficient

## Abstract

The research in storage and conversion of energy is an everlasting process. The use of fuel cells is very tempting but up to now there are still several conceptual challenges to overcome. Especially, the requirement of liquid water causes difficulties due to the temperature limit. Therefore, imidazoles and triazoles are increasingly investigated in a manifold of experimental and theoretical publications as they are both very promising in overcoming this problem. Recently, triazoles were found to be superior to imidazoles in proton conduction. An ab-initio molecular dynamics simulation of pure triazole phases for investigating the behavior of both tautomer species of the triazole molecule has never been done. In this work, we investigate the structural and dynamical properties of two different solid phases and the liquid phase at two different temperatures. We are able to show how the distinct tautomers contribute to the mechanism of proton conduction, to compute dynamical properties of the four systems and to suggest a mechanism of reorientation in solid phase.

## 1. Introduction

Modern society demands an ongoing development in energy conversion, storage and overall efficiency. This is caused by a progressive increase in world population and the use of electronic devices, which is undeniably further growing in the next decades. The promising field of fuel cells might be able to solve this problem sufficiently [1,2,3,4,5,6], but up to now several conceptual challenges need to be overcome. Especially, the need of liquid water in many types of fuel cells leads to problems due to the temperature limit.

Recently, azoles—especially imidazoles [7,8,9,10,11] and triazoles [12,13,14,15,16,17,18]—achieved attention within the field of fuel cells. Being incorporated in polymers, they serve as proton conducting agents. Foundation of the easy incorporation of triazoles in polymers and other molecules is the click chemistry [19,20,21,22] from which the era of triazoles began. Another hot field of research in material science is the ionic liquids which are often imidazolium- [23,24,25] or triazolium-based [26,27,28,29]. The applications of ionic liquids range from green chemistry [30,31,32,33] over electrolyte materials [26,29,34], lubricant agents [35,36] to proton conduction in general [24,37].

In the field of proton conduction [38,39,40,41,42], many investigations have been made to characterize the properties of azoles experimentally [12,39,43] and in computer simulations [44,45]. Static calculations have the benefit of demanding little computational time. Nevertheless, they offer no insight into the dynamics of the system as they are by definition frozen to zero Kelvin. For many properties like proton transfer, diffusion coefficients, and dimer forming, molecular dynamics simulations have to be used. Especially, ab-initio molecular dynamics simulations offer a high level description of the motion of the atoms. Such a simulation can be done at finite temperatures and offers an ensemble averaging of properties alongside the trajectory. Another great benefit is the inherent dynamical environment effects within the simulation as solvation shells and π-π stacking are accessible for investigation.

In the literature, there are two mechanisms of proton conduction widely accepted. On the one hand, there is vehicle diffusion of the charge carrying molecule transporting the charge throughout the system. On the other hand, there is the Grotthus mechanism also often referred to as a “jump” style mechanism. Especially, for the second, an understanding on molecular scales is very important. Within our ab-initio molecular dynamics simulation, we have been able to understand the behavior more precisely than by static calculations. Furthermore, we present a suggestion to a mechanism which goes hand in hand with former literature and our new findings. Even more, we have been able to simulate and investigatethe newly found crystalline phase of pure 1,2,3-triazole [43]. In total, we have investigated liquid 1,2,3-triazole at two different temperatures (413 K and 313 K) and two different crystal structures namely an orthorhombic phase at 280 K and a monoclinic phase at 269 K.

When it comes to triazoles, lately the conduction of lithium ions received attention [46,47,48] which makes our research even more important as an understanding of the proton conducting properties can be utilized to investigate lithium diffusion from a different perspective.

Furthermore, it is important to note that 1,2,3-triazole molecules occur in two distinct tautomeric forms. The 1H-triazole carries the acidic hydrogen atom at one of the outer nitrogen atoms of the aromatic ring. The 2H-triazole instead carries the acidic hydrogen atom at the middle nitrogen atom making it symmetrical. This symmetry influences the behavior dramatically as the resulting dipole is 42 times bigger in case of the non symmetrical 1H-triazole [45]. This property is reflected in the tautomer ratio found for different temperatures as well as in gas phase [43,45,49]. The higher the temperature gets, the more the 2H-triazole (less polar) is dominating. Furthermore, both tautomeric forms can transform through intermolecular tautomerization or through intramolecular tunneling [45,50] into the counterpart which makes it even more challenging from a material engineering perspective. Which tautomer is dominant is also dependent on the polarity of the solvent as the polarity increases the ratio shifts toward to the high dipole 1H-triazole.

Our focus of investigation is the hydrogen bond network as it is the foundation of proton conduction. Which tautomeric form takes part in the hydrogen bond network is therefore crucial for understanding proton conduction on a molecular scale. Lately, triazoles implemented in polymers have been investigated [51] where the question arose how the two tautomers influence the reorientation during proton conduction. Within our simulation, we are able to show how 1H-triazole and 2H-triazole molecules take part in proton conduction.

Finally, we show that the dynamics of our simulated systems reproduces the diffusion coefficient of experimental data [43].

## 2. Results

### 2.1. Orientation in Liquid and Crystalline Phases

In the framework of proton conduction, liquid phases offer high mobility of the charge carrying molecules. As the triazole molecules are isotropically distributed throughout the system, they have—within viscosity limits—the ability to freely rotate and rearrange. Nevertheless there are certain orientations within the liquid phase which are highly energetically favored. As triazoles posses an aromatic ring and acidic hydrogen atoms, they are predestined to undergo π-π stacking. To investigate this, we defined the criterion for the presence of π-π stacking pattern as shown in Figure 1. Within this definition, we take into account the intermolecular distance of the mass centers of the aromatic ring (CoM) as criterion number one. The second criterion is the angle between the normal of the ring plane and the connection line between the two ring centers of mass.

With these two criteria, we have calculated a combined distribution function which shows directly which angles between two triazole molecules are preferred and at which distance these angles are favored. In Figure 2, we present this plot for the liquid phase at 313 K (graph a). As a higher temperature results in a higher mobility of the molecules, a similar plot emerges for the liquid phase at 413 K but only with less predominant peaks in the combined distribution function. There are two forms of π-π stacking namely parallel and T-shaped alignment. Parallel stacking refers to angles of 0° and 180°. T-shaped π-π stacking occurs at angles around 90°. It is important to note that the peaks in our plots around 90° also contain hydrogen bonding as our criterion does not distinguish between them. Nevertheless, angles close to 90° are the predominant pattern in liquid phase. Figure 2 also shows that in the liquid phase, parallel stacking is common as well but less pronounced. For the parallel alignment, it is expected that both molecules are not exactly on top of each other. Both aromatic rings are slightly negative and therefore repel each other. The acidic hydrogen instead carries a positive charge. Favored is therefore an alignment where the hydrogen atom of one molecule is close to the ring center of the second as positive and negative charge attract each other. In Figure 2, we show that for both solid phases—orthorhombic at 280 K (graph b) and monoclinic at 269 K (graph c)—the T-shaped conformation reduces drastically (90°) and the parallel conformation is dominating. Undergoing the transition from liquid phase to an orthorhombic crystalline phase the former 500 pm distance at 90° is reduced to 370 pm which denotes that the hydrogen bonding is highly increased. Using the ring centers of mass as a criterion, the range from 350 to 500 pm distance is reasonable for hydrogen bonding. The smaller the distance between the two centers of mass, the better the hydrogen atoms are shared between the molecules and the stronger hydrogen bonded they are. For the monoclinic phase, this pattern shifts to 500 pm which leads to a more loosely bonded hydrogen network. These findings will be discussed more deeply for both phases in the next section.

In Figure 3, snapshots taken from the corresponding simulations of both crystalline phases are shown. The order within the crystals reflects nicely the findings of π-π stacking presented in Figure 2. As one can see in the graphs c and d, the 1H-tautomers form a wire structure with the acidic hydrogen pointing at a vacant nitrogen of their neighboring molecules (red marked nitrogen atoms in Figure 3), whereas the 2H-triazole (blue marked nitrogen atoms in graphs c and d in Figure 3) is pointing with its acidic hydrogen towards the vacant nitrogen atom within the wire.

### 2.2. H-Network and Role of Tautomers

For the conduction of protons, there are in literature two mechanisms widely accepted. On the one hand, there is the vehicle diffusion of whole molecules transporting the positive charge through the system. On the other hand, there is the Grotthuss mechanism where protons are transported in a concerted mechanism from one side of a conduction wire of molecules to the other side throughout a “jump” like mechanism. For the Grotthuss mechanism, the reorientation of the molecules after the proton “jump” is necessary. In liquid phase, this is easily achieved by rotation. In solid phase, nevertheless, the out of plane rotation is normally sterically hindered. 1,2,3-triazole has two tautomeric forms meaning both forms can have different or identical roles in the Grotthuss mechanism and reorientation. Therefore, it is crucial to have an understanding of how the two different tautomeric forms interplay in proton conduction. It has to be emphasized that the 2H-tautomer carries the acidic hydrogen on the middle nitrogen atom which makes it symmetrical. However, the 1H-tautomer carries it on an outer nitrogen atom. To investigate their role for proton conduction, we have calculated combined distribution functions. As the first criterion, we have chosen the bond length between the acidic hydrogen and the corresponding nitrogen it is attached to (intramolecular). The second criterion is the hydrogen bond distance to the nitrogen of other triazole molecules (intermolecular). If both distances are the same, the acidic hydrogen atom is shared equally between the two nitrogen atoms. In Figure 4, this is denoted by a red line. The closer the peaks are to this line, the stronger the hydrogens bonded the two observed species are. We have split our investigation into how favored it is for 1H-triazoles to hydrogen bond to other 1H-triazoles and how favored it is for 2H-triazole to hydrogen bond to other 1H-triazoles. Both are predominant pattern in solid phase. 1,2,3-triazole molecules have two vacant nitrogen atoms meaning hydrogen bonding can occur at both. To distinguish between them, we have calculated four different combined distribution functions which are shown in Figure 4. As one can see, the bond length between nitrogen and hydrogen atoms is oscillating around 100 pm The hydrogen bond distance, in contrast, shows always one predominant peak but a broad range of low occurrence at higher distances (also accounting for second layers of molecules at very high distances). We observed that there are two preferred pathways in hydrogen bonding for 1,2,3-triazoles. On the one hand, the 1H-triazoles are strongly hydrogen bonded to the outer nitrogen atoms of other 1H-triazoles (graph a in Figure 4). On the other hand, the 2H-triazoles are strongly hydrogen bonded to the middle nitrogen atom of 1H-triazole molecules (graph b in Figure 4). If proton transfer occurs, it is therefore most likely to take place in these two orientations. Furthermore, our investigation has shown that the two other possibilities (graphs c and d in Figure 4) are very unlikely for hydrogen bonding as distances of more than 250 pm are to distant for proton transfer. Summarizing this means that both tautomeric forms have distinct roles in the hydrogen network and furthermore in proton conduction itself. As discussed in the previous section, the wire is formed by 1H-triazoles which form a potential proton conducting tunnel and the 2H-triazoles pointing at the nitrogen atoms within the wire. Therefore, it is reasonable to assume that the proton conduction takes place along the red wire shown in Figure 3. If a proton approaches one side of the wire, the positive charge can be transported very fast within the Grotthuss mechanism to the other side. This results in a reverse situation of protonation compared to before the proton conduction. Now every outer nitrogen which was vacant before has a hydrogen atom attached to it and vice versa. If the reservoir of protons stays at the same position, this wire would now be blocked in this direction and a reorientation has to take place. A possible reorientation mechanism will therefore be discussed in the next section.

For the liquid phases, there was no distinct preference found for hydrogen atoms being donated to specific nitrogen atoms which can be explained by the higher freedom in motion.

### 2.3. Reorientation Mechanism

Within the Grotthuss picture of conduction, an elementary hopping step has to be followed by a reorientation step, which brings the system back into a configuration so that the subsequent hopping step can happen again. In many systems, this reorientation step is realized as a rotation of the molecule which has hosted the charge carrier, but such a step turns out to be sterically hindered for our case of 1,2,3-triazole in the solid state. Only very limited rotational movement is possible at ambient temperatures, according to our molecular dynamics trajectories.

However, our simulations have shown an alternative mechanism for returning the triazole system into the original protonation configuration (i.e., the same as prior to the hopping step). This mechanism is outlined in Figure 5. Step 1 describes the initial hopping process, which is followed by two specific proton exchange processes (steps 2 and 3 in Figure 5). Each of these exchange processes involve the concerted motion of two protons in one intermolecular (step 2) and one intramolecular (step 3) tautomerization reaction.

The tautomerization 2H:1H ↔ 1H:2H in step 2 has been studied computationally in the gas phase by Rauhut [45], who found a reaction barrier of about 40 kJ/mol.

This particular tautomerization was claimed to occur via a slight rotation of the two involved triazole molecules (see Figure 6).

We have analyzed our molecular dynamics trajectories to validate this rotational motion by means of a specific combined distribution function. The two variables of this distribution function are the angles between the intermolecular center-of-mass vector of two adjacent triazoles and the respective N-H bond of the 1H- and the 2H-triazole (see Figure 7).

The combined distribution function exhibits its most prominent feature at an angle of 0° for the 2H-triazole and 70° for the 1H triazole, which corresponds to intermolecular geometries as sketched in Figure 5. The width of this feature is about ±15°, which describes weak rotational fluctuations around that geometry.

More interesting, however, are the remaining peaks in the combined distribution function. The peak at 10°/60° represents the stable orientation with both triazoles rotated by 1/5 of a full rotation, but with the same proton orientation as plotted in Figure 5. In turn, the three connected features around 50–60°/50–60° correspond to transient configurations of the dimer which resemble the orientations sketched in Figure 6. These orientations are only moderately stable, meaning that they are observed frequently in the trajectory, but they are by an order of magnitude less stable than the dominant orientation.

It is of interest to show the difference of the hydrogen bonds in the proton conducting wire and outside of it. Therefore we have used the hydrogen bond distance in the wire as the first criterion and the hydrogen bond distance pointing at the wire as the second criterion for combined distribution functions shown in Figure 8. It can be seen that the hydrogen bond distance is 20 to 25 pm shorter inside the wire. This means the hydrogen bond network it stronger interconnected along the wire.

To get back to the original protonation configuration the third and last step of our mechanism from Figure 5 has to take place. One possibility is the intramolecular tunneling of the acidic hydrogen atom to a neighboring nitrogen atom to retain the starting configuration. Rahut [45] found this energy barrier for this internal tautomerization in gas phase to be at 98 kJ/mol. This is a rather high energy barrier but for the crystalline structure the result is energetically much more favored than in gas phase which might lower the real energy barrier drastically. Furthermore, for the orthorhombic phase there is a one to one ratio between both tautomers which offers a second possible mechanism. Instead of the necessity for both acidic hydrogen atoms to tunnel we suggest a surrogate mechanism which matches to the rotational freedom presented in Figure 7. The detailed mechanism is presented in Figure 9. Due to being more loosely connected to other triazole neighbors as seen in Figure 8, the first step in Figure 9 (from a to b) is most likely to be induced by the triazole outside the wire structure (upper molecule). Once it starts rotating, it might induce the rotation of the 1H-triazole within the wire (lower molecule) as well. In step b the acidic proton is donated forming two charged species. By rotating back in step c and back donation in step d the original configuration of hydrogen atoms is achieved. Back rotation in step e leads to the equilibrium structure before the proton conduction happened (structure 1 in Figure 5).

For the monoclinic phase, there arises the problem that the ratio of 1H-triazole to 2H-triazole is 2:1. This means that this last step therefore needs in half of the cases a double intramolecular hydrogen atom tunneling which does not mean it is impossible but could also explain the lower proton conductivity found in experiment [43] as this might happen more seldom. As stated at the beginning of this section, this is our suggestion to a reorientation mechanism but for us the most promising so far when it comes to solid 1,2,3-triazole phases.

### 2.4. Diffusion Coefficients

We have computed diffusion coefficients in both liquid phases via the root mean square displacements based on our ab-initio molecular dynamics trajectories, listed in Table 1. Both the molecular triazole and the proton diffusion coefficients are shown; the latter has been computed using the relative coordinate of the acidic proton with respect to the triazole molecule to which it was bonded initially. For each diffusion value, we also report the statistical variance in our (small) set of protons/triazoles within the simulation box.

It turns out that the vehicle diffusion is between one (for T = 313 K) and two (for T = 413 K) orders of magnitude larger than the proton diffusion, and that the increase in temperature from T = 313 K to T = 413 K results in an increase by one (vehicle) and two (proton) orders of magnitude. It should be noted here that these diffusion values are determined for the neutral species; the proton mobility may increase considerably in the presence of excess protons (i.e., under the typical conditions of a proton fuel cell).

The proton diffusion coefficient refers to the hoping diffusion experimentally determined by Pulst et al. [43]. The vehicle diffusion can be compared to the complete diffusion determined by Pulst et al. subtracted by the hopping diffusion. For the liquid phase at 313 K, both coefficients are in good agreement with the experimental results [43]. For 413 K, no experimental data is available but one order of magnitude difference compared to 313 K seems to be reasonable. Within ab-initio molecular dynamics simulation timescales it is reasonable that there is no diffusion determinable which leads to zero diffusion coefficient in solid phases.

## 3. Discussion

1,2,3-triazole is a very versatile molecule as it can be easily incorporated into polymers by click chemistry [19,20,21,22], adjusts itself by tautomerization according to the polarity of the solvent or surrounding (high polarity 1H-triazole, low polarity 2H-triazole) [45,50], and as a small molecule it offers high diffusivity [43]. For applications in fuel cells [2,6], it is crucial to know as much as possible about the properties of triazole molecules.

With our results, we have made a big step towards fully understanding the behavior of 1,2,3-triazoles. First of all, it is interesting that the temperature dependence of the proton diffusion constant is stronger than that of the vehicle diffusion constant. This indicates a larger enthalpic barrier for the diffusion, which is consistent with the intuitive understanding that breaking a chemical bond requires more energy than just a molecular displacement in a liquid. A clear future perspective is the simulation of proton diffusion in the presence of excess protons, which we expect to significantly reduce the hopping barrier and thus increase the diffusion constant. In general, nitrogen-based molecules offer a huge potential for lithium and hydrogen conduction. Our findings go hand in hand with former projects of ours. When we investigated imidazole-based polymers [52] with NMR methods we saw great potential in proton conduction and storage originating from the nitrogen-based imidazole part of the polymer.

Furthermore, we have figured out what the role of the two tautomeric forms is when it comes to proton conduction or the hydrogen bond network in general. Whereas the 1H-triazole is very likely to form wire structures and therefore improve the proton conductivity, the 2H-triazole acts rather like a supporter for returning to the original protonation configuration. This is very crucial when it comes to incorporation in polymers or other molecules as a bad ratio between the tautomers can hinder the proton conductivity [51].

We have been able to show that in solid phase the π-π stacking is of parallel nature whereas in liquid phase the T-shaped π-π stacking is the predominant pattern.

Our simulations show a mechanism for returning to the original protonation configuration and is backed up by literature [45]. Starting from our findings it is also possible to use our mechanism to understand the proton dynamics of 1,2,3-triazoles [12,50] and azoles in general in various environments like incorporated in membranes or polymers and their applications [12,13,14,15,16,17,18]. When we investigated the tautomerism of lithium 1,2,3-triazolate [53], we have not thought of a mechanism found within this work. Not only triazoles but amide and imide compounds in general offer huge potential. In a former project of ours, we investigated the lithium conduction which amides and imides are offering [54]. Therefore, our mechanism may be also applicable for transport of lithium ions [46,47,48] which is subject to future projects.

## 4. Materials and Methods

The ab-initio molecular dynamics simulations were calculated by the use of the CP2K software package [55,56]. The module Quickstep [57] was incorporated within the density functional theory as well as the molecularly optimized basis set (DZVP-MOLOPT-SR-GTH) [58] which is offering a good compromise between accuracy and simulation speed. In every simulation we utilized a time step of 0.5 fs. Nose–Hoover thermostat [59,60], the GTH-BLYP functional [61,62,63,64], DFTD3 dispersion correction [65] as well as the orbital transformation method of VandeVondele and Hutter [66] have been used. For every system, 10 ps of equilibration have been done. The systems themselves are further described in Table 2.

For the solid systems, crystal structures served as a starting configuration [43] and have been optimized geometrically. For the liquid phases, force field molecular dynamics within the Lammps software [67] and AMBER force field [68] have been simulated for the starting structures. The tautomer ratio of the liquid phase at 313 K is taken from Mauret [49]. The liquid phase at 413 K has the purpose to serve as a model system at elevated temperature and therefore was also chosen to have the same tautomer ratio as the liquid phase at 313 K.

The analysis of the trajectories has been done by the use of the TRAVIS software [69]. For the images, ChemDraw, Xmgrace, and Mathematica [70] have been used.

## Figures and Tables

**Figure 1 molecules-25-05722-f001:**
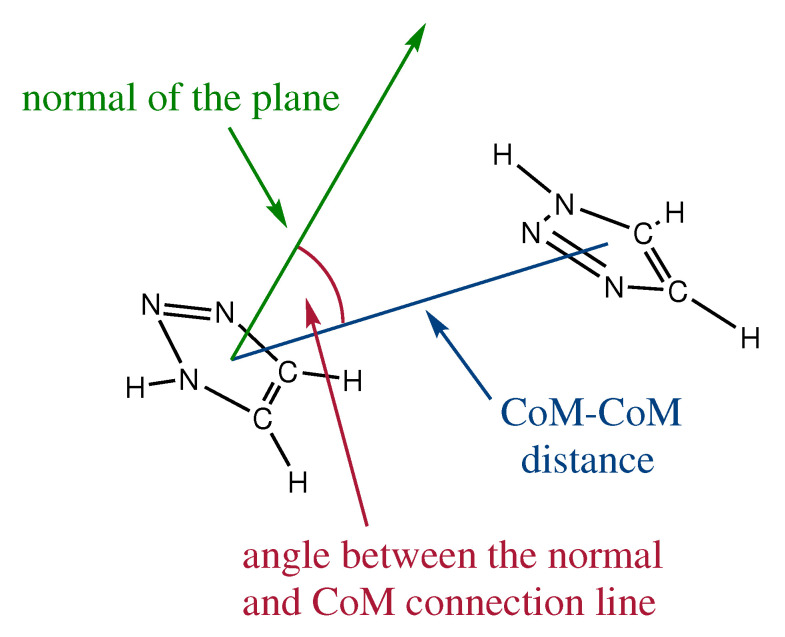
Definition of the criterion for π-π stacking. If the angle between the normal of the plane and the line between the two ring center of masses is zero than both molecules are parallel to and above each other. If the angle is 90°, it refers to T-shaped π-π stacking or a hydrogen bond.

**Figure 2 molecules-25-05722-f002:**
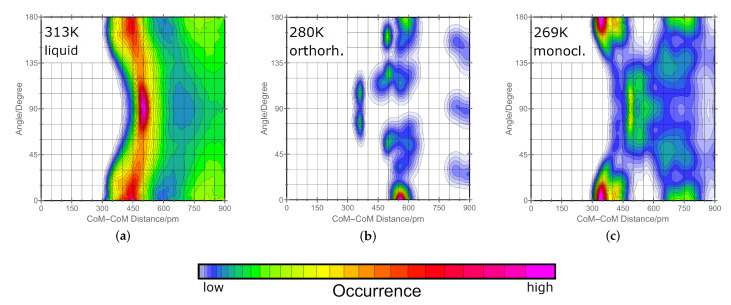
Combined distribution function of the distance between the ring mass centers of two triazole molecules and the angle between this connection line and the normal of the ring plane of the reference triazole molecule as defined in Figure 1. The graph (**a**) shows the orientations in liquid phase at 313 K. In the center (**b**), the corresponding graph for the orthorhombic phase at 280 K is shown and on the right (**c**), the orientations within the monoclinic phase at 269 K is presented. Lowermost, the color scheme for the qualitative probability of occurrence is given which holds for all the graphs above.

**Figure 3 molecules-25-05722-f003:**
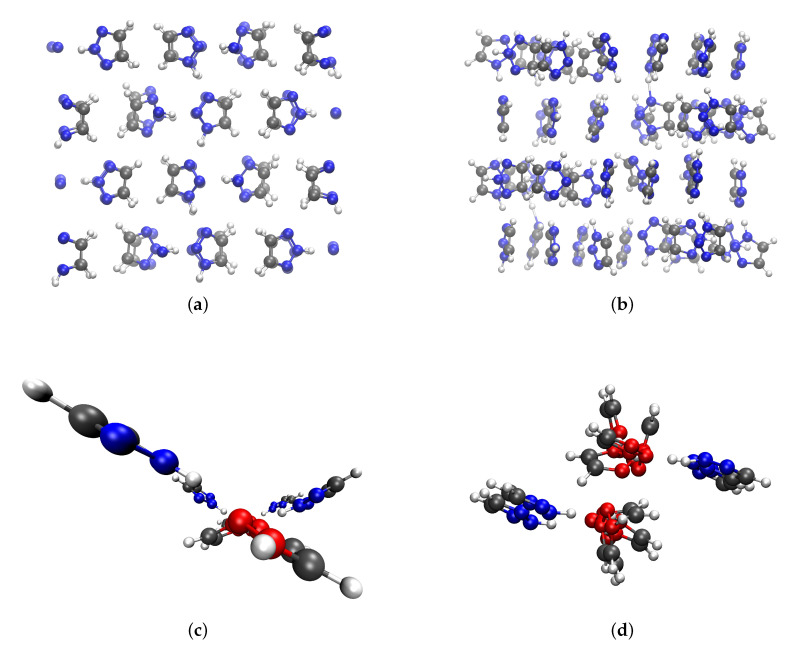
Graph (**a**) shows a snapshot taken from the simulation of the orthorhombic phase at 280 K. Below (graph (**c**)), the side view of a cutout of the crystal structure is shown. The graphs of the right side (**b**,**d**) show the same perspective but for the monoclinic phase at 269 K. In the lowermost graphs, nitrogen atoms of the 2H-triazole are shown in blue whereas the nitrogen atoms of the 1H-triazole are shown in red to distinguish between the two tautomers.

**Figure 4 molecules-25-05722-f004:**
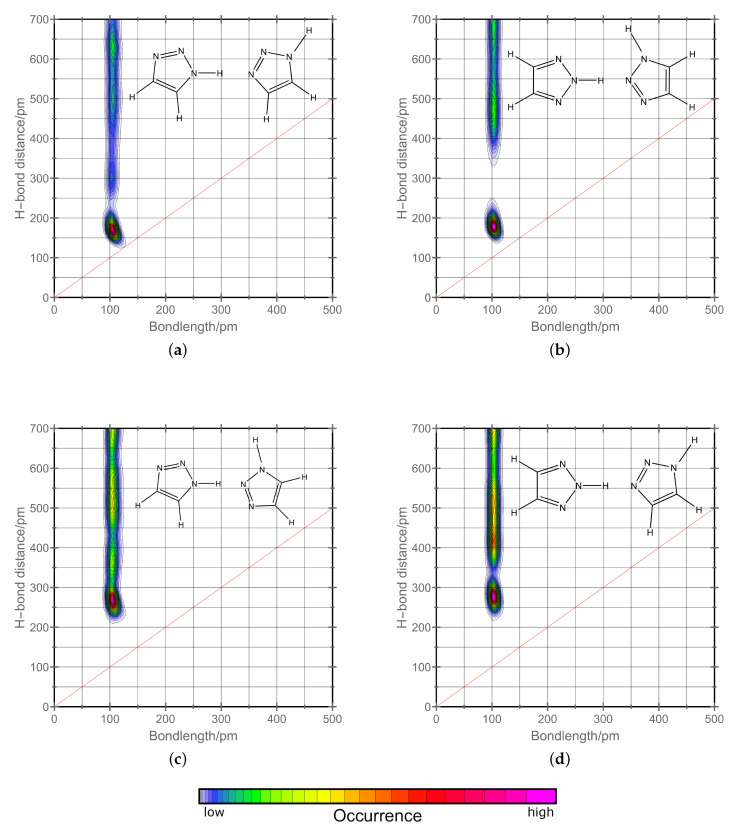
Combined distribution functions of the intramolecular nitrogen to hydrogen bond distance and the intermolecular distance of the acidic hydrogen to vacant nitrogen atoms in the monoclinic phase at 269 K. On the left side hydrogen bonding between two 1H-triazole molecules (**a**,**c**) and on the right side hydrogen bonding between 2H- and 1H-triazole molecules is shown (**b**,**d**). Qualitatively, the same results are observed for the orthorhombic phase at 280 K. The red line shows positions where both distance criteria have equal values. Lowermost, the color scheme for the qualitative probability of occurrence is given which holds for all the graphs above.

**Figure 5 molecules-25-05722-f005:**
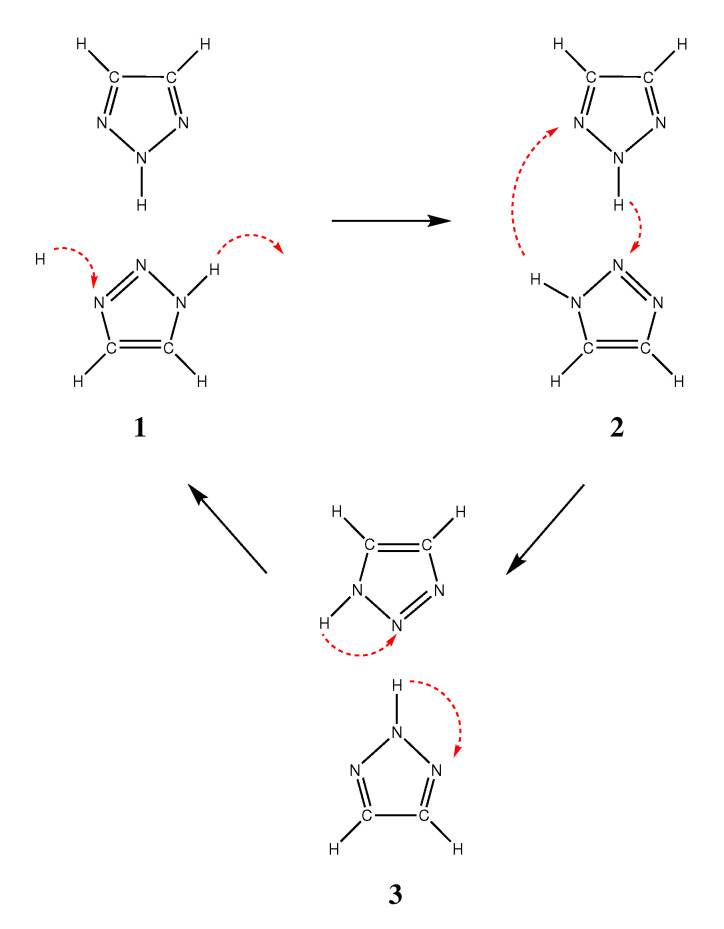
Suggested mechanism for the reorientation as a mandatory step within the Grotthuss mechanism. The first step shows the proton conduction mechanism along the built-in wire structure of the crystalline phases. Step two shows a concerted mechanism which is already discussed in literature [45] and is shown in detail in Figure 6. The third step is tunneling of the protons to their energetically more favorable starting configuration as it is the most stable within the crystal structure.

**Figure 6 molecules-25-05722-f006:**
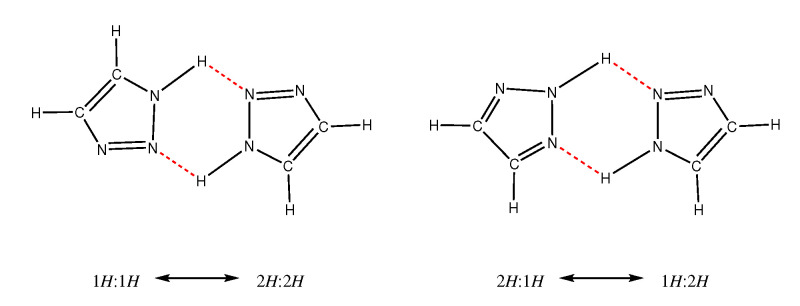
Two concerted mechanisms for tautomerization of triazoles. Rauhut [45] investigated these two mechanisms in gas phase with a reaction barrier of 40 kJ/mol.

**Figure 7 molecules-25-05722-f007:**
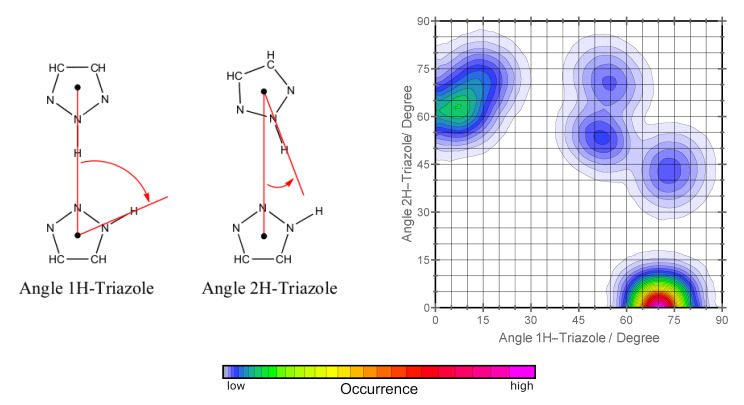
On the left side, the two angle criteria for the calculated combined distribution function are defined. On the right side, the calculated combined distribution function for the orthorhombic phase at 280 K is shown. Qualitatively, the monoclinic phase at 269 K shows the same behavior and was there not shown here. In the graph two predominant angle patterns within the crystals can be seen. The transition angles are seen with low occurrence. Lowermost the color scheme for the qualitative probability of occurrence is given.

**Figure 8 molecules-25-05722-f008:**
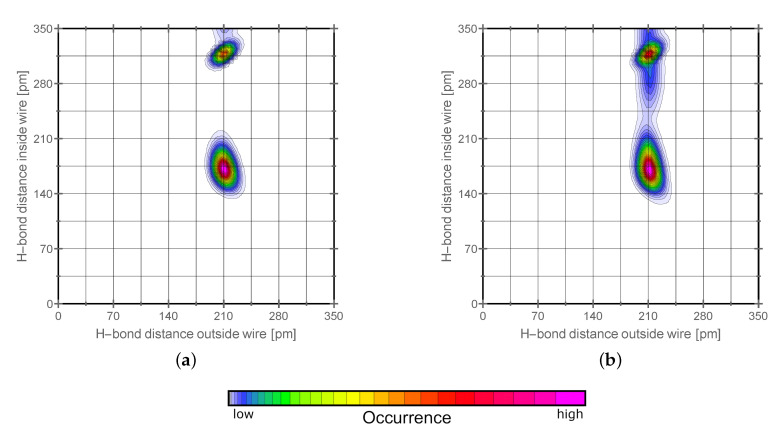
Shown are combined distribution functions for the orthorhombic phase left (graph (**a**)) and the monoclinic phase right (graph (**b**)). The criteria are the hydrogen bond distance along the wire plotted against the same distance outside the wire structure. Lowermost, the color scheme for the qualitative probability of occurrence is given which holds for all the graphs above.

**Figure 9 molecules-25-05722-f009:**
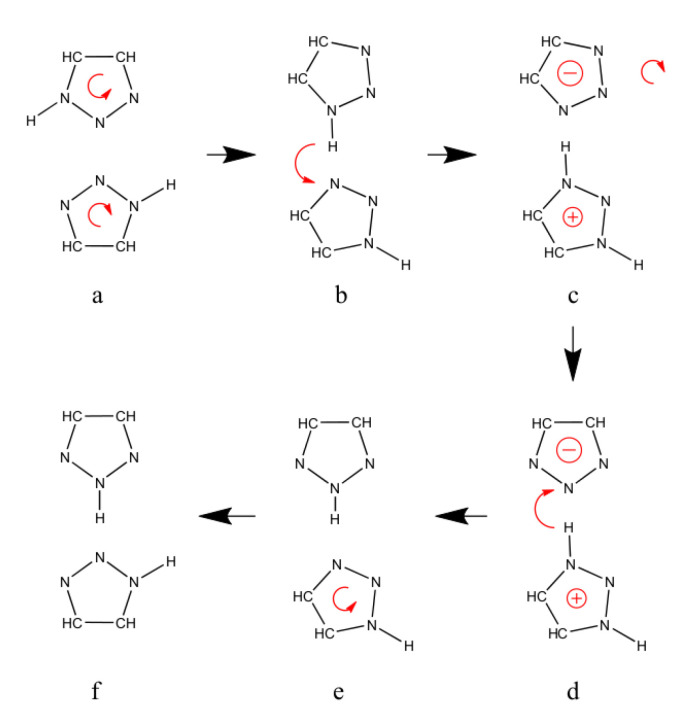
Detailed description of the third step of Figure 5. Step (**a**) is the starting configuration which undergoes a rotation resulting in the orientation of step (**b**). Proton donation leads to two charged species seen in step (**c**). Afterwards the negative charged species rotates backwards and gets its hydrogen back (step (**d**)). The lower triazole rotates back in step (**e**) and as a result, step (**f**) equals the configuration of step 1 of Figure 5.

**Table 1 molecules-25-05722-t001:** Computed diffusion coefficients via the root mean square displacement based on our ab-initio molecular dynamics trajectories for both liquid phases. D${}_{\mathrm{veh}}$ denotes the center of mass-based diffusion coefficient of triazole molecules and D${}_{\mathrm{diff}}$ denotes the independent movement of the acidic hydrogen. The latter has been calculated by using the relative coordinate of the acidic hydrogen with respect to center of mass of the triazole molecule it originally was bond to. The error bars represent the statistical variance obtained from the ensemble of particles.

Phase	Temperature	D${}_{\mathrm{veh}}$/$\frac{{\mathrm{cm}}^{2}}{s}$	D${}_{\mathrm{diff}}$/$\frac{{\mathrm{cm}}^{2}}{s}$
liquid	413 K	(1.2 ± 2.8) $\times \;10^{-5}$	(1.6 ± 1.6) $\times \;10^{-6}$
liquid	313 K	(2.0 ± 3.1) $\times \;10^{-6}$	(3.3 ± 6.8) $\times \;10^{-8}$

**Table 2 molecules-25-05722-t002:** Computational setup of all produced ab-initio molecular dynamics trajectories.

Trajectory	269 K	280 K	313 K	413 K
No. triazoles	48	32	32	32
Tautomer ratio 1H:2H	32:16	16:16	5:27	5:27
Physical time/ps	75	148	205	101
Boxvector x/Å	19.726	18.662	18.3291	18.3291
Boxvector y/Å	10.910	19.327	18.3291	18.3291
Boxvector z/Å	18.572	7.396	9.1646	9.1646
Crystal system	monoclinic	orthorhombic		
	β = 93.6085∘

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
