# Peer review of "ab-Initio Study of Hydrogen Bond Networks in 1,2,3-Triazole Phases"

_molecules, 2020, doi:10.3390/molecules25235722_

Round 1
Reviewer 1 Report
The authors have studied the proton transfer in triazoles using AIMD method and statistically sampled CDFs. The manuscript may be accepted in present form but a few minor queries/suggestions may be considered :
- Do authors find any evidence for charged triazoles in step c and d or Fig 9?
- The variance is of larger amplitude than the diffusion coefficient itself which is perplexing
- The authors may emphasize on finite temperature, dynamical environment effect, and ensemble averaging over static approaches in the Introduction.
- Is it possible to show a continous transfer from one intermediate to other through intermediate in the CDF of Fig 7 (b) ?
Author Response
Dear referee,
we would like to thank you for the fast reviewing process of our manuscript. We believe that we have addressed all of the remaining issues|either by adapting the manuscript text, or by discussing the issue.
Please find attached our response to the referee report. We marked the original statements of the referees by gray boxes, and we emphasize the changes in the manuscript text by sans serif font.
We hope that our manuscript is in a suitable form for publication now.
Yours sincerely,
Christopher Peschel

Reviewer 2 Report
The authors have studied proton conduction in triazoles at different temperatures. While the work seems to have the merit to be published in the journal of Molecules, the grammar in the manuscript is difficult to follow. As a whole, the English in the manuscript needs to be improved. Here are a few example sentences that I find difficult to follow or that are grammatically wrong.
Lines 34-35: For proton conduction there is the vehicle diffusion and Grotthus mechanism widely accepted and especially for the second an understanding on molecular scales is very important. Lines 40-41: In total we looked at 3 different phases at 4 different temperatures namely two liquid phases, an orthorhombic and a monoclinic crystalline phase.
Lines 77-79:
To distinguish the two forms of p-p stacking it is important to note that an angle of 0◦ respectively 180◦ (which is due to the definition just the other side of the ring) corresponds to parallel stacking of two molecules.
There are many more!
Here are some general suggestions to the authors. Using the present form of the manuscript, I am not able to provide any solid scientific suggestions (since it is taking more time to understand each paragraph!).
- The last paragraph of the introduction needs to be properly organized. It is very difficult to follow what the authors are aiming to study in this work. Probably, they are interested in studying proton conduction, but the title is suggesting that they are interested in H-bonding networks. The introduction does not imply the same.
- It would be nice to divide the figures into (a), (b), (c), etc. panels and discuss them separately. This suggestion applies to most of the figures.
Based on the computational details provided by the authors, here is my concern.
- It is surprising to see that the authors were able to distinguish the results performed at 269 and 290 K. Can the authors show the changes in the temperature profiles of these systems as a function of the simulation time? In the reviewer's experience, temperature fluctuations of around 30 K are expected (with a small number of atoms that the authors have considered). Distinguishing the results differed by 10 K does not seem to be correct.
Author Response
Dear referee,
we would like to thank you for the fast reviewing process of our manuscript. We believe that we have addressed all of the remaining issues|either by adapting the manuscript text, or by discussing the issue.
Please nd attached our response to the referee report. We marked the original statements of the referees by gray boxes, and we emphasize the changes in the manuscript text by sans serif font.
We hope that our manuscript is in a suitable form for publication now.
Yours sincerely,
Christopher Peschel

Reviewer 3 Report
This work is devoted to a comprehensive study of ab-initio molecular dynamics calculation in 1,2,3-triazole phases. The authors have carried out an investigation of 3 different phases at 4 different temperatures namely two liquid phases. They have made an important step to an understanding of 1,2,3-triazole behaviour. I convince, that this manuscript is suitable for publication in Molecules.
Several minor questions/comments:
1. How the theory level for calculation depends on the represented results. What is happening if DZVP basis set is changed to another one(for example, TZVP). A short rationale for choosing a theory level is necessary to add.
2. The discussion part is written too briefly. Adding some information on correlation with experimental and previous calculations will give some extra significance to the article.
3. The main focus of the present study is focused on the calculation of molecular geometry parameters in the condensed phase. Is it possible to estimate some energetic parameters of molecular interaction using combining Molecular Dynamics and Quantum Chemistry methods?
Author Response

(The authors gave the same response as above.)

Round 2
Reviewer 2 Report
I am satisfied with the authors' modifications. The manuscript can be accepted in its present form.